

# Total Global Solar Radiation Estimation with Relative Humidity and Air Temperature Extremes in Ireland and Holland

**Can Ekici,[a, b], Ismail Teke [a]**

[a]Yildiz Technical University, Department of Mechanical Engineering, Istanbul, Turkiye.
[b]Turkish Standard Institution, Gebze Calibration Laboratory, Kocaeli, Turkiye.
canekici@gmail.com

**Abstract**. Solar radiation is the earth's primary energy source for all biochemical and physical activities. Accurate knowledge of the solar radiation is important in engineering applications. This study aimed to calibrate some of the existing models in the literature for estimating daily total global solar radiation parameter using available measuring records (maximum and minimum air temperatures) and new models were developed based on maximum and minimum air temperatures, relative humidity and relative humidity extremes. Applicability of the Hargreaves model, Allen model, Bristow-Campbell model and Chen model were evaluated for computing the daily total solar global radiation, the geographical and meteorological data of Irish and Dutch cities were used. Meteorological data were taken from Royal Netherlands Meteorological Institute and Irish Meteorological Service. The models were compared on the basis of error tests which were mean percentage error (MPE), mean bias error (MBE), root mean square error (RMSE) and Nash-Sutcliffe equation (NSE). And, monthly MPE errors were given for each model. This study proposed new estimation models which were based on daily average relative humidity, relative humidity extremes and temperature extremes. Error analyses were applied to these models and results were given in the study.

**Keywords**: solar radiation; temperature; relative humidity; daily total global solar radiation; model comparison; Ireland; Holland; meteorological models; model validation

## 1   Introduction

Solar energy is the principal energy source for the processes such as biological, chemical and physical activities. Accurate knowledge of the solar radiation is important for many applications; simulations and modellings, architectural design, solar energy systems. There are many meteorological stations those measure basic meteorological parameters; but not all of them measure the global radiation in the worldwide. Sometimes, measurement of the solar radiation



cannot be available due to the equipment's cost, maintenance and calibration requirements in

developing countries. There are several empirical models in the literature to estimate the global

radiation using various parameters (Chen *et. al.,* 2004; Menges *et. al,* 2006).

Solar energy is an energy source, which is clean, renewable and domestic and solar energy has

high importance (Menges *et. al,* 2006). Without knowledge of solar radiation, it is impossible to

design solar energy systems. Estimation models are widely used when solar radiation is not

measured and available, these models help to obtain solar radiation value.

Amount of the solar radiation that received to the globe can change due to variables such as the

time of day and the season, and the prevailing atmospheric conditions... In the northern

hemisphere, the greatest amount of radiation is received in the location that is situated between 15

40   ºN and 35 ºN latitudes, for example Egypt. The next place which receive greatest amount of

radiation is between 15 ºN and the equator which includes Central America. Countries located

between the latitudes 35 ºN and 45 ºN, such as Spain and Turkey, show significant seasonal

variations resulting in less radiation received. The least favorable locations are situated beyond 45

44   ºN receive the least amount of direct radiation; such as Ireland, England, Norway, Holland and

Sweden. Approximately half of the radiation arrives at the surface as diffuse radiation, because

there may be frequent heavy cloud cover in the atmosphere (Armstorng *et. al,* 2010).

One of the main purposes of this study is the validation of the several models in the literature;

those use the difference between maximum, minimum air temperatures, to estimate daily total

global radiation in the cities of Ireland and Netherlands. These cities are Dublin, Eindhoven,

Groningen, Maastricht, Rotterdam and Twente. The study suggests new estimation models for the

prediction of the solar radiation. In this study, meteorological data for the cities were taken from

Royal Netherlands Meteorological Institute and Irish Meteorological Service database and used

for validation of the models. In the last years, calibration and metrology knowledge were

developed; new methodologies were submitted by commissions like Euramet. So, it is thought the

new data of meteorology institutes are more accurate and traceable. It has been thought that; the

measurement's reliability is higher in the data which have been recorded in recent past.

Meteorological parameters were taken between 2008 and first half of 2016.

Met Éireann, the Irish National Meteorological Service, is a line division of the Department of the

Environment, Community and Local Government. It is the leading provider of weather

information and related services for Ireland.

The Royal Netherlands Meteorological Institute (KNMI) is the Dutch national weather service.

Primary tasks of KNMI are weather forecasting, and monitoring of weather, climate, air quality

and seismic activity. KNMI is also the national research and information centre for meteorology,

climate, air quality, and seismology. KNMI focuses on monitoring and warning for risks with an

atmospheric or seismic origin. In addition, KNMI offers advice and strategy prospects for both

acute and future dangers. In order to improve future advice and therefore reach risk reduction, we

actively seek to learn from past events.

**2    Some of the Main Mathematical Formulas about the Solar Radiation**

Mathematical formulas about solar radiation, which were used in this study, are given in this part

of the paper.

The plane of rotation of the earth around the sun is called the ecliptic plane. The rotation axis of

the earth is called polar axis. The earth's rotation and the position of the earth axis causes diurnal

and seasonal changes in solar radiation. The angle between the sun and the equatorial plane of the

earth is different in every day of the year. This angle is called the solar declination angle; δ (Iqbal,

1983).


The solar declination angle's mathematical formula can be seen in equation 1. J is the calendar

77    day in this equation with J = 1 on January 1 and J = 365 (or 366 during leap years) on December

31 (Campbell *et. al.,* 1998).

$sin\delta = 0.39785 * \sin[278.97 + 0,9856J + 1.9165 * \sin(356.6 + 0,9856J)]$                (1)

Sunrise hour angle can be seen in equation 2. Here, $\omega_s$ is the sunrise angle; ø is the latitude of the

site (Iqbal, 1983).

$\omega_s = cos^{-1}[-\tan\text{ø} * tan\delta]$                (2)

Reciprocal of the square of the radius vector of the earth is called the eccentricity correction factor

of the earth's orbit, $E_o$. In many engineering applications, this factor can be expressed very simple.

The simple expression of the eccentricity factor can be seen in equation 3 (Iqbal, 1983).

$E_0 = 1 + 0.033 * cos[(\frac{2\pi*J}{365})]$                (3)

Mathematical equations are developed to determine the irradiation at various surface orientations

and for different time periods. Daily extraterrestrial radiation is shown in equation 4 (Iqbal, 1983).

$I_{sc}$ is the solar constant and it is equal to 4.921 MJ/day.m$^2$ (Menges *et. al,* 2006).

$H_0 = \frac{24}{\pi} * I_{sc} * E_0 * sin\text{ø} * sin\delta * [(\frac{\pi}{180}) * \omega_s - tan\omega_s]$                (4)

**3    Model Description**

*3.1  Hargreaves Model*

Hargreaves et al. (1985) suggested a simple method to estimate global solar radiation; the

expression can be seen in equation 5. "a" and "b" are the empirical coefficients. In this study,

Hargreaves model was used to predict daily total global solar radiation in Irish and Dutch cities.





$T_{max}$ can be taken as the daily maximum air temperature and $T_{min}$ is the daily minimum air

temperature. H is the daily total global solar radiation.

$$\frac{H}{H_0} = a * (T_{max} - T_{min})^{0.5} + b \tag{5}$$

*3.2 Allen Model*

Allen (1997) reported a self-calibrating model to estimate mean monthly global solar radiation,

which is the function of the mean monthly maximum and minimum temperatures. The model can

be seen in equation 6. In this study, this model was processed to estimate daily total global solar

radiation in the cities of Ireland and Netherlands.

$$\frac{H}{H_0} = a * (T_{max} - T_{min})^{0.5} \tag{6}$$

Also, "a" is an empirical coefficient, and it has been suggested as a mathematical expression, which

is the function ratio of the atmospheric pressure at site (P, kPa) and at sea level ($P_0$, 101.3 kPa) in

literature. The mathematical expression can be seen in equation 7. $K_{ra}$ value can be taken 0.17 for

interior regions, and 0.20 for coastal regions (Meza, 2000).

$$a = K_{ra} * \left(\frac{P}{P_0}\right)^{0.5} \tag{7}$$

*3.3 Bristow-Campbell Model*

Bristow and Campbell (1984) suggested a relationship between daily solar radiation as a function

of daily extraterrestrial radiation and the difference between maximum and minimum air

temperatures. The relationship can be seen in equation 8 and "a", "b" and "c" are the empirical

coefficients.

$$\frac{H}{H_0} = a * [1 - \exp(-b\Delta T^c)] \tag{8}$$



*3.4  Chen Model*

Chen et al. (2004) presented the model in equation 9.

$\frac{H}{H_0} = a * ln(T_{max} - T_{min}) + b$ (9)

*3.5  New Models Suggested in This Study*

Three models based on daily temperature extremes and daily average relative humidity are

suggested in the study. The models are shown in Eq. 10 and Eq. 11. *RH* is the relative humidity,

"a", "b", "c", "d" and "e" are the empirical coefficients. The $H_0$ value is calculated using the daily

parameters. The usage and explanations of these parameters are given in the previous sections.

Models will be used to calculate total daily global solar radiation values. In this study, the reason

why the period is selected on a daily basis is due to the importance of daily meteorological

estimations. It is also thought that there may be instantaneous changes in shorter time periods.

$\frac{H}{H_0} = a\left(\frac{RH}{100}\right) + b(T_{max} - T_{min})^{0.5} + c(T_{max} - T_{min}) + d\left(\frac{RH}{100}\right)(T_{max} - T_{min})^{0.5} + e$ (10)

$\frac{H}{H_0} = a \cdot [1 - \exp(-\Delta T^b)] + c \cdot RH$ (11)

Daily relative humidity extremes were used to estimate solar radiation in this study. Two models

were proposed for estimation the daily solar radiation related to relative humidity extremes. One

of the models use the saturation vapor pressure, the ratio between daily maximum relative humidity

and daily minimum relative humidity and the daily temperature extremes. Other model is based

on temperature extremes, relative humidity ratio and the relative humidity. $RH_{max}$ is the daily

measured maximum relative humidity, $RH_{min}$ is minimum relative humidity, $e_s$ is the saturation

vapor pressure at daily average temperature. The models are given in Eq. 12 and Eq. 13.

Calculation of $e_s$ is shown in Eq. 14. $T_{avg}$ is daily average air temperature.



$\quad \frac{H}{H_0} = a \cdot [1 - \exp(\{e_s \cdot (T_{max} - T_{min})^{0.5}\}^b)] + c \cdot \frac{RH_{min}}{RH_{max}}$ (12)

$\quad \frac{H}{H_0} = a \cdot [1 - \exp(\{T_{max} - T_{min}\}^{0.5b})] + c \cdot (T_{max} - T_{min})^{0.5} \cdot \frac{RH_{min}}{RH_{max}} + d \cdot (T_{max} - T_{min})^{0.5}$ (13)

$\quad e_s = 0.6108 \cdot \left[ \exp\left( \frac{17.27 \cdot T_{avg}}{T_{avg} + 237.3} \right) \right]$ (14)

Empirical coefficients of the models for the cities and performance of the models can be seen in

the next sections of the study.

## 142   **4   Climatic Data**

Daily climatic data for the Irish and Dutch cities were taken from meteorological public authorities

of Ireland and Netherlands; Royal Netherlands Meteorological Institute and Irish Meteorological

Service. Dublin, Eindhoven, Rotterdam, Groningen, Maastricht and Twente's daily meteorological

data were used in the study. Locations and altitudes of the meteorological stations are given in

Table 1.

The meteorological dataset is selected on a daily basis. These meteorological data belong to the

period between 2008 and July 2016. Maximum and minimum temperatures, daily total global solar

radiation, average daily relative humidity, daily maximum and minimum relative humidity values,

daily average temperature values were taken from meteorological stations. Extraterrestrial solar

radiation values were obtained by calculation. With the help of this data obtained from

meteorological stations, the models in the literature have been calibrated and new models have

been developed.

**Table 1** Location and altitude information of the meteorological stations

| Station name | Latitude | Longtitude | Altitude |
|---|---|---|---|
| Dublin | 53.423º | -6,238º | 71 m |
| Eindhoven | 51.451º | 5.377º | 22.6 m |
| Groningen | 53.125º | 6.585º | 5.2 m |
| Rotterdam | 51.962º | 4.447º | -4.3 m |
| Maastricht | 50.906º | 5.762º | 114.3 m |





| Twente | 52.274º | 6.891º | 34.8 m |

**5   Methods of Comparison and Model Evaluation**

Performances of the models were evaluated on the basis of mean percentage error (MPE), mean

bias error (MBE) and root mean square error (RMSE). MPE, MBE and RMSE are given in the

equation 15, 16 and 17. $H_{i,m}$ is the $i$th measured value, $H_{i,c}$ is the $i$th calculated value and $N$ is the

total number of observations (Menges *et. al,* 2006). RMSE gives information about the short term

performance of the correlations by using a term-by-term comparison of the deviations between the

observed and calculated values. MBE presents the systematic error or bias and provides

information on the long-term performance, positive value of MBE shows an over-estimate and

negative value gives an under-estimate by the model. Values of MPE are calculated from the actual

differences between calculated and measured values, and give the percentage errors of the

correlation (Almorox, 2011). When MBE converges to zero, it is the ideal performance for the

model, while a low value of RMSE and low MPE are desirable (Iqbal, 1983).

$MPE = \frac{1}{N}\sum_{i=1}^{n}\left[\frac{H_{i,c}-H_{i,m}}{H_{i,m}}\right] \cdot 100$ (15)

$MBE = \frac{\sum_{i=1}^{n} H_{i,c}-H_{i,m}}{N}$ (16)

$RMSE = \sqrt{\left|\frac{\sum_{i=1}^{n}(H_{i,c}-H_{i,m})^2}{N}\right|}$ (17)

The Nash-Sutcliffe equation is also an evaluation method. A model is more efficient when NSE is

closer to 1. The equation is shown in equation 18. $\bar{H}_m$ is the mean measured global radiation

(Menges *et. al,* 2006).

$NSE = 1 - \frac{\sum_{i=1}^{n}(H_{i,m}-H_{i,c})^2}{\sum_{i=1}^{n}(H_{i,m}-\bar{H}_m)^2}$ (18)



## 6    Results and Discussions

Solar radiation data can give useful information in the design and for studies about the solar energy

systems, agricultural processes, etc. In the literature, there are empirical models to estimate global

solar radiation. These models can be suitable tools if the parameters can be calibrated for the

different locations. In this study, some of the models in the literature were calibrated for Irish and

Dutch cities to estimate daily total global solar radiation. Also, five new models were presented in

this study and these models were validated with the meteorological data of Ireland and Holland.

Validation of the models were performed with MPE, MBE, RMSE and NSE methods and given

in the rest of the study.

### 6.1  Hargreaves Model

In equation 5, Hargreaves model can be seen. *a* and *b* are the empirical coefficients. In this study,

these empirical coefficients to estimate daily total global solar radiation in Irish and Dutch cities

are found and given in Table 2. The coefficients were derived by using MATLAB R2015a and

Minitab Statistical Software.

**Table 2** Empirical coefficients for Hargreaves model

| Location | "a" coefficient | "b" coefficient |
| --- | --- | --- |
| Dublin | 0.1472 | -0.01362 |
| Eindhoven | 0.1777 | -0.1336 |
| Groningen | 0.1716 | -0.1004 |
| Maastricht | 0.1983 | -0.1739 |
| Rotterdam | 0.1814 | -0.1045 |
| Twente | 0.1609 | -0.09308 |

MPE, MBE, RMSE error analyze methods have been applied on the model. And, NSE value has

been calculated via Excel 2013. The values are shown in Table 3. Also, mean percentage error for

the every month is given in Table 3.





NSE values show good fit between calculated and measured values for Dutch cities, but for Dublin

it is worse. Maximum average MPE values of Hargreaves model is around 20 percent. It may be

acceptable, but in some months MPE values are higher than others; for instance winter months. In

Dutch cities the errors in April, in Dublin the error in May are more satisfactory.

**Table 3** Error analyses of the Hargreaves model

| Location | | Monthly MPE | Whole of the model | |
|---|---|---|---|---|
| Dublin | January | -38.061 | MBE | 0.02 |
| | February | -20.832 | RMSE | 3.22 |
| | March | -14.052 | MPE | -22.18 |
| | April | -11.314 | NSE | 0.80 |
| | May | -8.364 | | |
| | June | -14.341 | | |
| | July | -17.631 | | |
| | August | -15.353 | | |
| | September | -12.452 | | |
| | October | -41.353 | | |
| | November | -31.560 | | |
| | December | -41.494 | | |
| Eindhoven | January | -23.704 | MBE | 0.21 |
| | February | -21.700 | RMSE | 2.89 |
| | March | -12.285 | MPE | -17.06 |
| | April | -8.681 | NSE | 0.86 |
| | May | -14.863 | | |
| | June | -13.962 | | |
| | July | -12.756 | | |
| | August | -12.048 | | |
| | September | -15.123 | | |
| | October | -14.361 | | |
| | November | -27.165 | | |
| | December | -25.968 | | |
| Groningen | January | -34.742 | MBE | 0.35 |
| | February | -18.449 | RMSE | 3.07 |
| | March | -13.620 | MPE | -18.89 |
| | April | -8.559 | NSE | 0.844 |
| | May | -14.581 | | |
| | June | -15.865 | | |
| | July | -11.197 | | |
| | August | -13.164 | | |
| | September | -17.897 | | |
| | October | -25.430 | | |
| | November | -28.266 | | |
| | December | -25.459 | | |
| Maastricht | January | -26.767 | MBE | 0.22 |
| | February | -22.254 | RMSE | 2.94 |
| | March | -15.592 | MPE | -20.46 |
| | April | -11.914 | NSE | 0.86 |
| | May | -16.599 | | |
| | June | -17.894 | | |



| | | | |
|---|---|---|---|
| | July | -15.036 | |
| | August | -13.171 | |
| | September | -14.800 | |
| | October | -20.179 | |
| | November | -27.354 | |
| | December | -45.167 | |
| **Rotterdam** | January | -32.303 | MBE -0.01 |
| | February | -29.201 | RMSE 3.19 |
| | March | -13.401 | MPE -19.78 |
| | April | -7.483 | NSE 0.84 |
| | May | -13.943 | |
| | June | -11.204 | |
| | July | -10.658 | |
| | August | -10.848 | |
| | September | -15.424 | |
| | October | -25.473 | |
| | November | -34.461 | |
| | December | -34.136 | |
| **Twente** | January | -25.681 | MBE 0.22 |
| | February | -22.185 | RMSE 3.06 |
| | March | -12.945 | MPE -18.32 |
| | April | -10.124 | NSE 0.84 |
| | May | -18.467 | |
| | June | -15.587 | |
| | July | -14.841 | |
| | August | -15.441 | |
| | September | -16.108 | |
| | October | -18.629 | |
| | November | -24.909 | |
| | December | -25.613 | |

## 6.2 Allen Model

Allen model was applied for the estimation of the daily solar global radiation in Irish and Dutch

cities. Empirical coefficient "a" was found by MS Office Excel 2013, coefficients can be seen in

Table 4. Error analyses of the Allen method's application is seen in Table 5. NSE value is seen

usable in the table. But some of the monthly MPE values are higher than Hargreaves Model. In

November and December, there are higher deviations between the predicted and observed values.

**Table 4** Empirical coefficients for Allen model

| Location | "a" coefficient |
|---|---|
| Dublin | 0.1418 |
| Eindhoven | 0.1291 |
| Groningen | 0.1335 |



| | |
|---|---|
| Maastricht | 0.1317 |
| Rotterdam | 0.1403 |
| Twente | 0.1266 |

**Table 5** Error analyses of Allen model

| Location | | Monthly MPE | Whole of the model | |
|---|---|---|---|---|
| **Dublin** | January | -38.507 | MBE | -0,02 |
| | February | -21.173 | RMSE | 3,24 |
| | March | -13.993 | MPE | -22,19 |
| | April | -11.070 | NSE | 0,80 |
| | May | -8.075 | | |
| | June | -13.786 | | |
| | July | -17.252 | | |
| | August | -15.013 | | |
| | September | -12.390 | | |
| | October | -41.641 | | |
| | November | -32.044 | | |
| | December | -41.987 | | |
| **Eindhoven** | January | -36.444 | MBE | -0.24 |
| | February | -30.261 | RMSE | 3.11 |
| | March | -13.986 | MPE | -19.20 |
| | April | -3.287 | NSE | 0.84 |
| | May | -12.722 | | |
| | June | -8.941 | | |
| | July | -10.254 | | |
| | August | -8.647 | | |
| | September | -11.853 | | |
| | October | -16.067 | | |
| | November | -38.415 | | |
| | December | -40.400 | | |
| **Groningen** | January | -45.674 | MBE | -0.23 |
| | February | -25.194 | RMSE | 3.21 |
| | March | -14.749 | MPE | -20.46 |
| | April | -4.751 | NSE | 0.83 |
| | May | -11.963 | | |
| | June | -11.910 | | |
| | July | -8.204 | | |
| | August | -9.629 | | |
| | September | -15.218 | | |
| | October | -27.360 | | |
| | November | -37.143 | | |
| | December | -34.041 | | |
| **Maastricht** | January | -45.347 | MBE | -0.38 |
| | February | -36.250 | RMSE | 3.29 |
| | March | -17.914 | MPE | -24.22 |
| | April | -6.889 | NSE | 0.82 |
| | May | -13.008 | | |
| | June | -11.612 | | |
| | July | -10.268 | | |
| | August | -7.599 | | |



| | | | | |
|---|---|---|---|---|
| | September | -11.910 | | |
| | October | -22.050 | | |
| | November | -42.285 | | |
| | December | -66.486 | | |
| Rotterdam | January | -41.692 | MBE | -0.34 |
| | February | -35.084 | RMSE | 3.34 |
| | March | -13.571 | MPE | -21.32 |
| | April | -3.626 | NSE | 0.82 |
| | May | -12.121 | | |
| | June | -8.053 | | |
| | July | -8.785 | | |
| | August | -8.523 | | |
| | September | -13.645 | | |
| | October | -27.340 | | |
| | November | -41.934 | | |
| | December | -42.664 | | |
| Twente | January | -37.525 | MBE | -0.17 |
| | February | -29.122 | RMSE | 3.18 |
| | March | -14.001 | MPE | -19.99 |
| | April | -5.542 | NSE | 0.83 |
| | May | -14.504 | | |
| | June | -10.647 | | |
| | July | -11.175 | | |
| | August | -12.255 | | |
| | September | -13.571 | | |
| | October | -20.498 | | |
| | November | -34.708 | | |
| | December | -37.001 | | |

*6.3  Bristow-Campbell Model*

Bristow-Campbell model's equation can be seen in equation 8. "a", "b" and "c" are the empirical

coefficients. They are shown in Table 6 for the estimation of the daily total global solar radiation

in Ireland and Holland.

**Table 6** Empirical coefficients for Bristow-Campbell model

| Location | "a" coefficient | "b" coefficient | "c" coefficient |
|---|---|---|---|
| Dublin | 1.991 | 0.5956 | 0.066 |
| Eindhoven | 1.260 | 0.9157 | 0.050 |
| Groningen | 1.644 | 0.7726 | 0.053 |
| Maastricht | 0.975 | 1.0940 | 0.051 |
| Rotterdam | 0.833 | 1.0690 | 0.075 |
| Twente | 2.523 | 0.7001 | 0.036 |

MBE, MPE, RMSE and NSE error analyses were applied to the model. These analyses and

monthly MPE analyses can be seen in Table 7. NSE value can be assumed as acceptable. Some of



the monthly MPE values do not give satisfaction for example in winter months. But for other

220     months; it can be said, the deviations are not too high.

**Table 7** Error analyses of Bristow-Campbell model

| Location | | Monthly MPE | Whole of the model | |
|---|---|---|---|---|
| Dublin | January | -37.256 | MBE | 0.03 |
| | February | -20.188 | RMSE | 3.22 |
| | March | -13.768 | MPE | -21.81 |
| | April | -11.149 | NSE | 0.80 |
| | May | -8.306 | | |
| | June | -14.489 | | |
| | July | -17.660 | | |
| | August | -15.368 | | |
| | September | -12.175 | | |
| | October | -40.657 | | |
| | November | -30.748 | | |
| | December | -40.631 | | |
| Eindhoven | January | -17.552 | MBE | 0.12 |
| | February | -16.621 | RMSE | 2.85 |
| | March | -10.278 | MPE | -13.86 |
| | April | -7.904 | NSE | 0.86 |
| | May | -13.163 | | |
| | June | -12.926 | | |
| | July | -12.312 | | |
| | August | -11.762 | | |
| | September | -13.168 | | |
| | October | -10.953 | | |
| | November | -21.150 | | |
| | December | -19.429 | | |
| Groningen | January | -32.0205 | MBE | 0.11 |
| | February | -16.228 | RMSE | 3.05 |
| | March | -12.427 | MPE | -17.65 |
| | April | -8.362 | NSE | 0.85 |
| | May | -14.272 | | |
| | June | -15.520 | | |
| | July | -10.861 | | |
| | August | -12.845 | | |
| | September | -17.167 | | |
| | October | -23.721 | | |
| | November | -25.838 | | |
| | December | -23.074 | | |
| Maastricht | January | -20.244 | MBE | 0.24 |
| | February | -17.202 | RMSE | 2.91 |
| | March | -12.745 | MPE | -17.65 |
| | April | -12.227 | NSE | 0.86 |
| | May | -16.204 | | |
| | June | -17.867 | | |
| | July | -14.500 | | |
| | August | -13.061 | | |





| | | | | |
|---|---|---|---|---|
| | September | -13.336 | | |
| | October | -17.095 | | |
| | November | -21.225 | | |
| | December | -37.184 | | |
| Rotterdam | January | -23.510 | MBE | 0.09 |
| | February | 23.125 | RMSE | 3.17 |
| | March | -11.610 | MPE | -17.02 |
| | April | -8.831 | NSE | 0.84 |
| | May | -13.676 | | |
| | June | -11.797 | | |
| | July | -10.555 | | |
| | August | -11.245 | | |
| | September | -15.239 | | |
| | October | -22.519 | | |
| | November | -27.176 | | |
| | December | -26.169 | | |
| Twente | January | -37.525 | MBE | -0.17 |
| | February | -29.122 | RMSE | 3.18 |
| | March | -14.001 | MPE | -19.99 |
| | April | -5.543 | NSE | 0.83 |
| | May | -14.505 | | |
| | June | -10.647 | | |
| | July | -11.175 | | |
| | August | -12.255 | | |
| | September | -13.571 | | |
| | October | -20.498 | | |
| | November | -34.708 | | |
| | December | -37.000 | | |

*6.4 Chen Model*

Chen model's empirical coefficients are seen in Table 8.

**Table 8** Empirical coefficients for Chen model

| Location | "a" coefficient | "b" coefficient |
|---|---|---|
| Dublin | 0.1841 | 0.0269 |
| Eindhoven | 0.2337 | -0.1014 |
| Groningen | 0.2168 | -0.0521 |
| Maastricht | 0.2695 | -0.1525 |
| Rotterdam | 0.2244 | -0.0464 |
| Twente | 0.2083 | -0.0539 |

MBE, MPE, RMSE and NSE error analyses can be seen in Table 9. Also, the monthly MPE

analysis is shown in table.

**Table 9** Error analyses of Chen model

| Location | Monthly MPE | Whole of the model |
|---|---|---|



| | Month | Value | Metric | Metric Value |
|---|---|---|---|---|
| **Dublin** | January | -36.680 | MBE | 0.01 |
| | February | -20.583 | RMSE | 3.25 |
| | March | -12.998 | MPE | -21.59 |
| | April | -9.803 | NSE | 0.80 |
| | May | -8.268 | | |
| | June | -14.297 | | |
| | July | -18.558 | | |
| | August | -16.293 | | |
| | September | -11.417 | | |
| | October | -39.952 | | |
| | November | -30.568 | | |
| | December | -40.496 | | |
| **Eindhoven** | January | -21.837 | MBE | 0.17 |
| | February | -20.914 | RMSE | 2.98 |
| | March | -14.049 | MPE | -16.83 |
| | April | -8.623 | NSE | 0.85 |
| | May | -14.085 | | |
| | June | -14.082 | | |
| | July | -13.721 | | |
| | August | -13.726 | | |
| | September | -16.830 | | |
| | October | -15.798 | | |
| | November | -26.121 | | |
| | December | -23.303 | | |
| **Groningen** | January | -32.959 | MBE | 0.09 |
| | February | -18.642 | RMSE | 3.15 |
| | March | -13.912 | MPE | -18.87 |
| | April | -8.518 | NSE | 0.84 |
| | May | -13.705 | | |
| | June | -16.450 | | |
| | July | -11.113 | | |
| | August | -13.522 | | |
| | September | -19.326 | | |
| | October | -27.241 | | |
| | November | -27.246 | | |
| | December | -24.593 | | |
| **Maastricht** | January | -20.563 | MBE | 0.37 |
| | February | -17.567 | RMSE | 3.05 |
| | March | -16.768 | MPE | -20.01 |
| | April | -12.301 | NSE | 0.85 |
| | May | -17.037 | | |
| | June | -20.378 | | |
| | July | -17.256 | | |
| | August | -15.340 | | |
| | September | -18.403 | | |
| | October | -23.050 | | |
| | November | -23.829 | | |
| | December | -39.248 | | |
| **Rotterdam** | January | -30.659 | MBE | -0.03 |
| | February | -29.140 | RMSE | 3.22 |
| | March | -14.228 | MPE | -19.65 |
| | April | -7.401 | NSE | 0.83 |
| | May | -13.046 | | |
| | June | -11.287 | | |
| | July | -10.569 | | |


| | | | | |
|---|---|---|---|---|
| | August | -11.742 | | |
| | September | -16.618 | | |
| | October | -26.414 | | |
| | November | -33.475 | | |
| | December | -32.638 | | |
| Twente | January | -23.901 | MBE | 0.18 |
| | February | -23.060 | RMSE | 3.17 |
| | March | -13.966 | MPE | -18.31 |
| | April | -10.164 | NSE | 0.83 |
| | May | -18.233 | | |
| | June | -15.642 | | |
| | July | -14.539 | | |
| | August | -16.079 | | |
| | September | -17.899 | | |
| | October | -20.544 | | |
| | November | -22.554 | | |
| | December | -23.844 | | |

*6.5  Ekici Models*

Three daily solar radiation estimation models are suggested in this study. They were shown in

Equation 10, 11, 12 and 13. There are empirical coefficients in the models. The empirical

coefficients of the models can be seen in Table 10. These coefficients are calculated by regression

analyses of Minitab 17 Statistical Software and MATLAB fitting toolboxes. In the table, Equation

10 is called as Ekici's Model 1, Equation 11 is Model 2 and Equation 12 and Equation    13    are

named as Model 3 and Model 4.

**Table 10** Empirical coefficients for Ekici models

| # | Location | "a" coefficient | "b" coefficient | "c" coefficient | "d" coefficient | "e" coefficient |
|---|---|---|---|---|---|---|
| | Dublin | -1.092 | -0.0333 | 0.009703 | 0.1331 | 1.007 |
| | Eindhoven | -1.224 | -0.1198 | 0.01446 | 0.2098 | 1.091 |
| *Model 1* | Groningen | -1.435 | -0.156 | 0.01554 | 0.2321 | 1.343 |
| *(Eq. 10)* | Maastricht | -1.433 | -0.2583 | 0.03107 | 0.2874 | 1.348 |
| | Rotterdam | -1.472 | -0.2572 | 0.03116 | 0.2803 | 1.413 |
| | Twente | -1.256 | -0.1483 | 0.02002 | 0.1801 | 1.216 |
| | Dublin | -0.4202 | 0.09728 | -0.007322 | | |
| | Eindhoven | -0.3242 | 0.1198 | -0.00599 | | |
| *Model 2* | Groningen | -0.4326 | 0.0931 | -0.007682 | | |
| *(Eq. 11)* | Maastricht | -0.350 | 0.1138 | -0.00647 | - | - |
| | Rotterdam | -0.4068 | 0.1047 | -0.007442 | | |
| | Twente | -0.3921 | 0.09542 | -0.007086 | | |
| | Dublin | -0.6164 | -0.02444 | -0.920 | | |
| *Model 3* | Eindhoven | -0.5782 | -0.01691 | -0.9104 | | |
| *(Eq. 12)* | Groningen | -0.6233 | -0.01365 | -0.9556 | - | - |
| | Maastricht | -0.5752 | 0.003312 | -0.9478 | | |



|  |  |  |  |  |  |  |
|---|---|---|---|---|---|---|
|  | Rotterdam | -0.6457 | -0.009491 | -1.026 |  |  |
|  | Twente | -0.5729 | -0.01314 | -0.9082 |  |  |
|  | Dublin | -0.1046 | 0.3166 | -0.21034 | 0.166 |  |
|  | Eindhoven | 4.47•10⁻⁶ | -2.000 | 0.130 | 0.202 |  |
| *Model 4* | Groningen | 0.001094 | 1.210 | -0.2093 | 0.2899 | - |
| *(Eq. 13)* | Maastricht | 0.210 | 0.520 | -0.1923 | 0.5897 |  |
|  | Rotterdam | 0.00081 | 1.256 | -0.2441 | 0.319 |  |
|  | Twente | 0.006525 | 0.9105 | -0.2017 | 0.2839 |  |

RMSE, MBE, MPE and NSE error analyses were executed to the application of the models that

are suggested in the study to estimate solar radiation of Irish and Dutch cities. The error values can

be seen in the Table 11. Error values can be seen as acceptable, monthly MPE values are also seen

as acceptable. For Dublin, in January, December and October, the monthly MPE values are higher

than the others. For Dutch cities, in May, the monthly values are seen higher than other months.

The correlation between the observed and the measured values (NSE) for all cities are seen

acceptable.

**Table 11** Error analyses of Ekici models

| Location |  | Monthly MPE | | | | | Whole of the model | | | |
|---|---|---|---|---|---|---|---|---|---|---|
|  |  | Model 1 | Model 2 | Model 3 | Model 4 |  | Model 1 | Model 2 | Model 3 | Model 4 |
| Dublin | January | -25.235 | -24.388 | -18.213 | -13.394 | MBE | 0.12 | 0.14 | -0.26 | -0.20 |
|  | February | -10.202 | -10.384 | -11.488 | -4.729 | RMSE | 2.87 | 2.88 | 3.04 | 2.85 |
|  | March | -11.597 | -11.098 | -10.927 | -6.530 | MPE | -15.61 | -15.60 | -12.17 | -10.57 |
|  | April | -11.708 | -11.104 | -11.396 | -9.094 | NSE | 0.84 | 0.84 | 0.82 | 0.84 |
|  | May | -10.182 | -10.663 | -10.092 | -7.244 |  |  |  |  |  |
|  | June | -15.929 | -16.458 | -10.480 | -13.134 |  |  |  |  |  |
|  | July | -15.513 | -16.528 | -8.728 | -12.087 |  |  |  |  |  |
|  | August | -13.247 | -13.997 | -8.500 | -10.298 |  |  |  |  |  |
|  | September | -5.481 | -5.320 | -2.284 | -3.650 |  |  |  |  |  |
|  | October | -26.453 | -26.050 | -19.642 | -21.148 |  |  |  |  |  |
|  | November | -17.868 | -17.478 | -14.118 | -10.018 |  |  |  |  |  |
|  | December | -23.569 | -23.641 | -19.324 | -15.885 |  |  |  |  |  |
| Eindhoven | January | -8.835 | -9.163 | -6.242 | -0.433 | MBE | 0.21 | 0.23 | 0.12 | -0.27 |
|  | February | -13.657 | -12.540 | -15.225 | -3.400 | RMSE | 2.50 | 2.52 | 2.67 | 2.56 |
|  | March | -12.550 | -11.735 | -19.983 | -5.134 | MPE | -9.94 | -10.20 | -9.74 | -4.23 |
|  | April | -11.340 | -11.690 | -14.066 | -6.359 | NSE | 0.89 | 0.89 | 0.88 | 0.89 |
|  | May | -15.829 | -16.826 | -17.411 | -9.980 |  |  |  |  |  |
|  | June | -14.657 | -15.341 | -12.688 | -8.924 |  |  |  |  |  |
|  | July | -11.137 | -12.053 | -10.107 | -7.627 |  |  |  |  |  |
|  | August | -7.655 | -7.965 | -5.326 | -4.727 |  |  |  |  |  |
|  | September | -4.628 | -4.582 | 0.683 | -1.414 |  |  |  |  |  |
|  | October | -1.345 | -1.563 | 2.193 | 3.127 |  |  |  |  |  |
|  | November | -9.766 | -10.257 | -10.474 | -4.589 |  |  |  |  |  |
|  | December | -6.660 | -7.570 | -5.699 | -0.796 |  |  |  |  |  |
| Gr... | January | -15.920 | -17.355 | -19.812 | -11.472 | MBE | 0.19 | 0.22 | 0.15 | -0.18 |



| Station | Month | | | | | Metric | | | | |
|---|---|---|---|---|---|---|---|---|---|---|
| | February | -8.471 | -9.072 | -14.571 | -3.804 | RMSE | 2.69 | 2.72 | 2.83 | 2.74 |
| | March | -12.085 | -11.751 | -19.692 | -8.338 | MPE | -11.41 | -12.06 | -12.69 | -7.95 |
| | April | -10.680 | -11.224 | -13.434 | -7.985 | NSE | 0.88 | 0.88 | 0.87 | 0.88 |
| | May | -18.449 | -19.006 | -18.554 | -12.480 | | | | | |
| | June | -19.135 | -19.683 | -16.147 | -13.248 | | | | | |
| | July | -10.085 | -10.637 | -8.251 | -7.276 | | | | | |
| | August | -7.770 | -8.009 | -4.920 | -6.443 | | | | | |
| | September | -8.914 | -8.849 | -4.650 | -6.274 | | | | | |
| | October | -9.194 | -10.796 | -9.728 | -8.948 | | | | | |
| | November | -6.440 | -8.270 | -11.528 | -6.733 | | | | | |
| | December | -8.368 | -8.743 | -8.266 | -1.505 | | | | | |
| Maastricht | January | -11.981 | -13.557 | -6.351 | -3.049 | MBE | 0.20 | 0.26 | 0.17 | -0.38 |
| | February | -12.894 | -13.262 | -13.523 | -5.014 | RMSE | 2.56 | 2.60 | 2.89 | 2.65 |
| | March | -15.778 | -16.126 | -22.315 | -9.260 | MPE | -12.49 | -13.71 | -12.37 | -6.44 |
| | April | -13.430 | -14.107 | -16.024 | -8.168 | NSE | 0.89 | 0.89 | 0.86 | 0.88 |
| | May | -15.524 | -17.091 | -18.371 | -10.377 | | | | | |
| | June | -15.283 | -16.430 | -15.796 | -9.899 | | | | | |
| | July | -11.854 | -13.351 | -13.047 | -6.925 | | | | | |
| | August | -9.867 | -10.356 | -12.796 | -5.931 | | | | | |
| | September | -5.210 | -5.871 | -4.380 | -0.843 | | | | | |
| | October | -6.255 | -7.507 | -2.746 | -0.367 | | | | | |
| | November | -11.456 | -12.673 | -8.681 | -3.417 | | | | | |
| | December | -19.431 | -23.383 | -12.587 | -13.317 | | | | | |
| Rotterdam | January | -12.753 | -14.002 | -19.495 | -12.373 | MBE | -0.10 | 0.14 | 0.15 | 0.12 |
| | February | -13.132 | -14.693 | -16.008 | -11.746 | RMSE | 2.80 | 2.83 | 3.03 | 2.87 |
| | March | -9.348 | -10.602 | -11.886 | -7.111 | MPE | -10.45 | -12.47 | -13.89 | -11.65 |
| | April | -5.673 | -7.933 | -8.921 | -8.512 | NSE | 0.87 | 0.87 | 0.85 | 0.87 |
| | May | -12.697 | -15.982 | -18.672 | -16.642 | | | | | |
| | June | -11.266 | -13.943 | -13.896 | -12.566 | | | | | |
| | July | -10.053 | -12.864 | -15.893 | -13.827 | | | | | |
| | August | -7.876 | -10.825 | -10.229 | -11.122 | | | | | |
| | September | -6.429 | -8.056 | -6.183 | -8.472 | | | | | |
| | October | -8.100 | -10.846 | -9.107 | -10.975 | | | | | |
| | November | -11.574 | -13.213 | -15.741 | -13.437 | | | | | |
| | December | -16.452 | -16.651 | -19.986 | -13.202 | | | | | |
| Twente | January | -10.432 | -8.949 | -10.447 | -2.942 | MBE | 0.21 | 0.20 | 0.12 | 0.10 |
| | February | -10.972 | -10.570 | -13.437 | -5.158 | RMSE | 2.55 | 2.56 | 2.62 | 2.56 |
| | March | -11.132 | -10.558 | -17.593 | -8.649 | MPE | -9.99 | -9.76 | -10.21 | -7.58 |
| | April | -12.212 | -12.455 | -14.283 | -12.194 | NSE | 0.89 | 0.89 | 0.89 | 0.89 |
| | May | -19.080 | -19.676 | -18.206 | -17.750 | | | | | |
| | June | -15.377 | -15.624 | -12.137 | -13.460 | | | | | |
| | July | -11.850 | -12.117 | -9.708 | -11.698 | | | | | |
| | August | -7.437 | -7.942 | -5.319 | -8.728 | | | | | |
| | September | -2.294 | -2.481 | 1.179 | -2.307 | | | | | |
| | October | 0.475 | -0.861 | 3.326 | 0.883 | | | | | |
| | November | -7.174 | -5.421 | -12.309 | -4.469 | | | | | |
| | December | -10.611 | -8.737 | -11.010 | -3.257 | | | | | |

A graphic showing the differences between the measured and calculated solar radiation values of

the models on daily basis for the month of February 2008 was drawn. This graphic is given in

Figure 1; it may be give idea about the models' daily trends.  If you look at the daily trends of the

models in the literature, it is seen that these models have more scattered errors. But in developed



models, it can be said that the errors are a little bit more closer to each other on daily basis. Since

it can be said that all models show the same tendency in general.

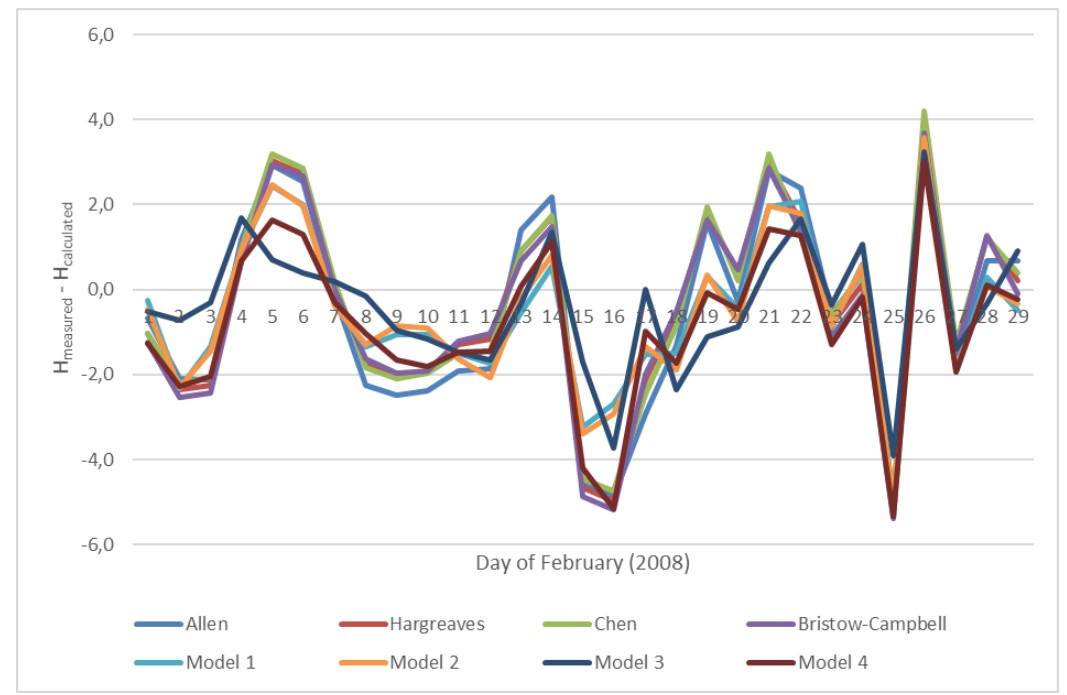

**Figure 1** Differences between measured and calculated daily total global solar radiation values in February 2008

**6    Conclusions**

Empirical models are usable tools to estimate global solar radiation, if the radiation parameters are

not available in the station. Main aim of this study is estimation of the daily total solar global

radiation values by using maximum and minimum daily air temperatures and daily average and

extreme relative humidity values. The daily data were taken from meteorological agencies of

Ireland and Holland. These data are daily total global solar radiation, daily average relative

humidity values, daily relative humidity extremes, daily minimum air temperatures and daily

maximum air temperatures. Data were selected between 2008 and 2016's first half. It is thought;

the recent measurements are more accurate and traceable.





Hargreaves, Allen, Bristow-Campbell and Chen models were applied to the cities for the prediction

of the daily total global solar radiation.

MBE and RMSE values explain the systematic errors of the models. When MBE value converges

to zero; the systematic error of the model decreases. It can be illustrated by bull's eye example. A

marksman wants to shot a bull from its eye. The bull's eye on the target represents the measured

solar radiation parameter we wish to estimate. If the marksman's aim is accurate, he/she scores a

bull's eye; on the other hand, the marksman misses the bull's eye by some distance. And the

marksman shoots the bull's eye repeatedly at the target, each time aiming at the bull's eye.  The

distance between the point clusters that shot by the marksman and the center of the eye explains

the mean bias error (Biemer *et. al.,* 2003). Hargreaves and Allen models have got good agreement

in mean bias errors for Dutch and Irish cities, but for Dublin the value of MBE is seen better than

other cities' values. The situation of Dublin about MBE values for Bristow-Campbell and Chen

models are seem similar as Hargreaves and Allen models. Allen Model's MBE values are greater

than other three models' MBE values. Ekici models' MBE values are closer to the MBE values of

other models. The greatest value of MBE in Ekici models is seen in Maastricht for Model 4. RMSE

values of all models are seen closer to each other, but in Ekici models RMSE values are a little bit

better than others. It can be said; the systematic errors of the models are similar, Ekici models'

values are a little bit lesser than others.

NSE is a method that indicates how well the plot of observed versus simulated data fits the line.

If NSE equals to 1, the model corresponds to a perfect match between modelled and observed data.

Nash-Sutcliffe error analyses were applied to the all models. All of the models' NSE values are

greater than 0.80. Ekici models in Eindhoven, Maastricht and Twente show best fits in the study

and have got the greatest NSE values.




Whole of the model mean percentage errors of models will be discussed. MPE values of Allen

model, Hargreaves model, Chen model and Bristow-Campbell model are seen closer to each other,

lay between -15 % ~ -20 %. The best value (-13.86 %) is seen in Eindhoven's Bristow-Campbell

model, the worst value (-24.22 %) is seen in Allen Model for Maastricht. Ekici models give better

performance in MPE analyses. Model 4 performs best in MPE analyses. The best performance is

seen in Eindhoven for Model 4. It is thought, the main reason of that situation is caused by using

more parameters than other Ekici models. Saturation vapor pressure is an extra parameter in Model

4 to describe solar radiation, which related to average air temperature.. In MPE analyses of this

study, all of Ekici models show better performances than other models those exist in the literature.

In monthly MPE analyses, Allen model has got higher errors than other models. Bristow-Campbell

model shows better monthly MPE performance than Chen model and Hargreaves model. In winter

297    months, models do not fit the measured values as well. It is thought; cloudy days affect to the

model performance in prediction of solar radiation with low accuracy. Monthly performances of

Ekici models are better than the models in literature. Best monthly MPE results are seen in Model

4.

In this study, four new models that are based on the relative humidity, relative humidity extremes

and the difference between maximum and minimum air temperatures were suggested. Model 1

and 3 gives good score in mean bias error. But all of the Ekici models' MBE and RMSE values

are closer to each other. NSE values are all of the Ekici models are similar. So it can be said; all

of the Ekici models show good agreement between calculated and measured values. All of the four

models give better scores in error analyses than the other models that exist in the literature for the

estimation of the Irish and Dutch cities' daily total solar global radiation.



**Conflict of interest**

The author declares that there is no conflict of interest regarding the publication of this article.

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
