# Peer review of "Total Global Solar Radiation Estimation with Relative Humidity and Air Temperature Extremes in Ireland and Holland"

_Geoscientific Instrumentation, Methods and Data Systems, 2017_

## Short Comment (SC1) · 19 Dec 2017

This article looks interesting for publishing. I think this article will be read a lot because solar radiation estimation is a very cited topic. However, professional English language editing process can be useful for manuscript.

---

## Author Comment (AC1) · 20 Dec 2017

Dear Yilmaz,

First of all, thank you for your valuable comments.

English language copy-editing for final revised paper is included in the APC. Copernicus declared this on their web site.

If our manuscript will be accepted for publishing, the article's last revision will be edited professionally.

Best regards.

---

## Short Comment (SC2) · 29 Dec 2017

Dear Author,

Thank you for your response.

---

## Referee Comment (RC1) · Anonymous Referee #1 · 6 Feb 2018

This article deals with new correlations to estimate the solar radiation from two standard weather data (air temperature and relative humidity). After presenting some available models in literature, the authors proposed new models. Results for some locations in Ireland and Netherlands show an improvement of the accuracy of the estimations. A lot of correlations were presented in previous articles [Menges (2006)]. The main contribution of this paper compared to previous publications comes from the association between air temperature and relative humidity to estimate solar radiation over daily or monthly periods. This articles has some paragraph that can be more synthetized. Some details can be removed, for instance the function of weather institutes (lines 58-67). Furthermore, results for some locations can be merged in a single table by putting

each location in one column (tables 3, 5, 7, 9). The conclusion brings some explanations about results of part 5. These explanations are needed to be written in part 5. Particularly, explanations about error estimations should be moved to part 5 with the tables of results. The explanation of over evaluation of solar radiation during clear days and lower estimation during cloudy days should be moved to the results page and developed. How they authors planned to address nebulosity in their model (see some approaches in literature) ? Figure 1 has to be completed with units on the axis (Units of the vertical axis (a priori MJ/d/m$^2$) ), axis position revised and data original location need to be specified. Some explanations about tables of part 5 can moreover be highlighted by adding a synthesis of the evolution of weather conditions over the studied month on a new figure. Evolution of relative deviation and pertinent error estimators should be added on another figure. All the studied locations have similar oceanic climate. It should be interesting to compare models for other climates (warmer as Mediterranean or tropical, dryer as continental). About the form, written English can be enhanced. Mathematical notations need to be written in italic in text.

---

## Author Comment (AC2) · 11 Feb 2018

First of all many thanks to the all reviewers for their precious comments.

Revised manuscript is given in supplements.

Response to the reviewer is given below.

Reviewer: Some details can be removed, for instance the function of weather institutes (lines 58- 67).

Answer: Details about meteorology institutes were removed from the manuscript.

[Figure]

Reviewer: Furthermore, results for some locations can be merged in a single table by putting each location in one column (tables 3, 5, 7, 9).

Answer: Table 3, 5, 7, 9 were merged in Table 3. Results of the all locations are given in one table.

Reviewer: The conclusion brings some explanations about results of part 5. These explanations are needed to be written in part 5. Particularly, explanations about error estimations should be moved to part 5 with the tables of results.

Answer: Explanations about error estimations given in the conclusions section were moved to section 5.

Reviewer: The explanation of over evaluation of solar radiation during clear days and lower estimation during cloudy days should be moved to the results page and developed.

Answer: Explanations about clear days and lower estimation days were moved to the end of Results and Discussion section.

Reviewer: How they authors planned to address nebulosity in their model (see some approaches in literature) ?

Answer: Ben Jemaa and others (2013) used nebulosity measurements to construct the cloud attenuation factor, which was then integrated to obtain the average total amount of monthly energy received. Daily temperature variations can reflect the cloud attenuation. Models given in the manuscript used the differences between maximum and minimum temperatures.

Reviewer: Figure 1 has to be completed with units on the axis (Units of the vertical axis (a priori MJ/d/m2 ) ), axis position revised and data original location need to be specified.

Answer: The data given in the Figure belongs to Eindhoven. Information on this was added to the text and Figure. Units of the vertical axis were added.

Reviewer: Some explanations about tables of part 5 can moreover be highlighted by adding a synthesis of the evolution of weather conditions over the studied month on a new figure. Evolution of relative deviation and pertinent error estimators should be added on another figure.

Answer: A new figure was added for weather conditions about given month. Comments were added about weather conditions.

Reviewer: All the studied locations have similar oceanic climate. It should be interesting to compare models for other climates (warmer as Mediterranean or tropical, dryer as continental).

Answer: Results for other climate types can be a new issue for a future study. Results of this study can be compared with applications in Mediterranean areas in a new study.

Reviewer: About the form, written English can be enhanced. Mathematical notations need to be written in italic in text.

Answer: English language copy-editing for final revised paper is included in the APC. Copernicus declared this on their web site. Mathematical notations were written in italic text.

Please also note the supplement to this comment:
https://www.geosci-instrum-method-data-syst-discuss.net/gi-2017-52/gi-2017-52-AC2-supplement.pdf

**Supplement:**

[revised manuscript text omitted]

---

## Referee Comment (RC2) · Anonymous Referee #2 · 14 Feb 2018

General comments:

The manuscript reports on a study focusing on the performance of five different parameterized models to estimate daily global solar radiation. Solar radiation is a major driver for atmospheric processes and life on the Earth. It may be pointwise measured in situ on the Earth's surface. However, the coverage of the measurement networks is limited, and the spatial resolution of the observations may not be good enough. Modelled estimates on global solar radiation, based on meteorological data more frequently available, are therefore highly welcome.

The manuscript reads well and is well-organized. However, it could benefit from lan-

guage checking.

Specific comments:

Page 1 Line 24: Introduction. The need for model estimates is well justified on the basis of the limited number of the ground-based instruments measuring in situ global solar radiation. How about satellite data: Are there any space-born data that could be used to give estimates on global solar radiation on a better spatial resolution globally? In which sense use of the estimates given by the parameterized models could be more convenient? The reader would appreciate if you could briefly discuss on that.

Page 3 Line 90: Please give the name of the quantity H. In SI system I think it is "radiant exposure". Sometimes it is called "dose" and the term "daily dose" is frequently used within, e.g., photobiology. In your study, you focus on the daily doses (total energy accumulated over the day). Could you please justify the choice? Why not an hourly value or a weekly or a monthly value?

Page 5 Line 98: I presume the temperature is in Kelvins. Please mention this to the reader.

Page 5 Line 100: What is "a self-calibrating model"? Please briefly explain the term.

Page 6 Line 118: Please explain the symbols used in the equation, even if they were exactly the same as in the previous equation.

Page 7 Line 156: Table 1. Longtitude -> Longitude.

Tables 2,4,6,8,10: You could consider presenting the data as a scatter plot. It is challenging for the reader to get a general view on the performance of the model just by looking at the numbers in the tables.

Page 10 Line 198: Winter months seem to be more difficult for the model. Any idea what could be the reason? Could you please discuss on the potential reasons?

Page 20 Line 250: What is "the models' daily trend"? Please specify.
Page 20 Fig. 1. Why is the Ekici model exluded? What are the Models 1-4? Any idea on the factors causing the fluctuation in the curve? The reader would appreciate a brief discussion on the potential factors. Perhaps you could look at least into the days with min and max deviation and try to trace how these days differ from the other days as regards their conditions.

---

## Author Comment (AC3) · 16 Feb 2018

Response to Reviewer 2: Reviewer: The manuscript reports on a study focusing on the performance of five different parameterized models to estimate daily global solar radiation. Solar radiation is a major driver for atmospheric processes and life on the Earth. It may be pointwise measured in situ on the Earth's surface. However, the coverage of the measurement networks is limited, and the spatial resolution of the observations may not be good enough. Modelled estimates on global solar radiation, based on meteorological data more frequently available, are therefore highly welcome. The manuscript reads well and is well-organized. However, it could benefit from language

checking.

Answer: Thank you very much for your precious views. English language copy-editing for final revised paper is included in the APC. Copernicus declared this on their web site. Language editing will be done professionally before it is published.

Specific comments: Page 1 Line 24: Introduction. The need for model estimates is well justified on the basis of the limited number of the ground-based instruments measuring in situ global solar radiation. How about satellite data: Are there any space-born data that could be used to give estimates on global solar radiation on a better spatial resolution globally? In which sense use of the estimates given by the parameterized models could be more convenient? The reader would appreciate if you could briefly discuss on that.

Answer: A paragraph about ground-based and satellite-based models was added to the Introduction Section. The paragraph is given below.

"Different types of models have been developed to estimate solar radiation when it is not measured (Gueymard et. al, 2008). There are several models in the literature, but a perfect model does not exist. A perfect model would be impossible due to measurement uncertainty and "true" solar irradiance cannot be determined theoretically (Gueymard et. al, 2008, Menges et. al, 2006). Ground-based statistical models show high performance. These models use one or more ground-based measurements as input parameters. However, there can be several errors in the estimations when using these models due to inaccurate data measured using un-calibrated and/or inaccurate instruments (Aksoy et. al, 2010). On the other hand, there are models in the literature that estimates ground level solar radiation using satellite data. Meteorological satellites provide observations of the atmospheric system. These satellite-based models can be divided in two categories: statistical approach based on relationship between satellite and ground data and a physical approach using radiative transfer models to express the relationship between satellite and ground measurements (Cano et. al, 2010). Validation of models based on satellite input data is much more complicated (Aksoy et. al, 2010). Temporary and spatial consistency questions are particularly annoying, as satellite data, while uniform, are usually sparse in time compared to surface observations. Spatial concerns are an even bigger problem, as surface observations are 'point' observations and satellite observations are spatially extended, even if at very high spatial resolution (Gueymard et. al, 2008)."

Page 3 Line 90: Please give the name of the quantity H. In SI system I think it is "radiant exposure". Sometimes it is called "dose" and the term "daily dose" is frequently used within, e.g., photobiology. In your study, you focus on the daily doses (total energy accumulated over the day). Could you please justify the choice? Why not an hourly value or a weekly or a monthly value? Answer: Name of the quantity H and introduction about units were added to manuscript as below. "The purpose of this study is modelling and reaching to the daily total global solar radiation. Its notation is H. It refers to total energy accumulated over the day on horizontal plane of the ground. It can be said that, this value is the total daily dose. Daily total global solar radiation and extraterrestrial solar radiation expresses in energy per square meter. Daily total solar global radiation is in MJ/(day.m2)." Hourly values of the solar radiation are complicated to estimate due to cloud motions. There can be many reasons; like measurement devices' response times, lower differences between hourly maximum and minimum air temperatures can cause the modelling results to be incorrect. Monthly values and annual values usually are mean averages. It is easy to reach annual and monthly average values. Daily values cannot be available in data sets and need to be calculated.

Page 5 Line 98: I presume the temperature is in Kelvins. Please mention this to the reader.

Answer: The sentence below was added to the manuscript. "Tmax and Tmin given in the models can be used in units of Celcius." The differences between maximum and minimum temperatures are important. So, it does not matter to take the temperatures in Kelvin or Celcius. But Tavg must be taken in Celcius. "Tavg is daily average air

temperature in Celcius." Was added to Section 3.5

Page 5 Line 100: What is "a self-calibrating model"? Please briefly explain the term.

Answer: A new sentence about self-calibration was added as below. The derivation of the coefficient a by the Equation 7 for regional stations allows that the model is self-calibrated.

Page 6 Line 118: Please explain the symbols used in the equation, even if they were exactly the same as in the previous equation.

Answer: Explanations about Chen model's symbols were added to the manuscript.

Page 7 Line 156: Table 1. Longtitude -> Longitude. Tables 2,4,6,8,10: You could consider presenting the data as a scatter plot. It is challenging for the reader to get a general view on the performance of the model just by looking at the numbers in the tables.

Answer: Longtitude were changed to longitude. Result tables of Hargreaves, Allen, Chen and Bristow-Campbell model combined into one table (Table 2 in new manuscript).

Page 10 Line 198: Winter months seem to be more difficult for the model. Any idea what could be the reason? Could you please discuss on the potential reasons?

Answer: A new paragraph was added to Section 6.5 about winter conditions as below. "In winter months, the weather conditions may be more complicated, as clouds, precipitation etc. Expression of the solar radiation with mathematical model becomes more difficult in cloudy and complicated weather conditions."

Page 20 Line 250: What is "the models' daily trend"? Please specify.

Answer: "the models' daily trends" words was changed to "daily error tendencies". These words can express the situation better.
Page 20 Fig. 1. Why is the Ekici model exluded? What are the Models 1-4? Any idea on the factors causing the fluctuation in the curve? The reader would appreciate a brief discussion on the potential factors. Perhaps you could look at least into the days with min and max deviation and try to trace how these days differ from the other days as regards their conditions.

Answer: Explanation about Models 1-4 was added for figure as below. "In the figure, Equation 10 is called as Model 1, Equation 11 is Model 2 and Equation 12 and Equation 13 are named as Model 3 and Model 4 for Ekici models." A new figure was added for weather conditions about given month. Discussions were added about weather conditions.